# SCORE-BASED CAUSAL DISCOVERY FROM HETEROGENEOUS DATA

## ABSTRACT

Causal discovery has witnessed significant progress over the past decades. Most algorithms in causal discovery consider a single domain with a fixed distribution. However, it is commonplace to encounter heterogeneous data (data from different domains with distribution shifts). Applying existing methods on such heterogeneous data may lead to spurious edges or incorrect directions in the learned graph. In this paper, we develop a novel score-based approach for causal discovery from heterogeneous data. Specifically, we propose a Multiple-Domain Score Search (MDSS) algorithm, which is guaranteed to find the correct graph skeleton asymptotically. Furthermore, benefiting from distribution shifts, MDSS enables the detection of more causal directions than previous algorithms designed for single domain data. The proposed MDSS can be readily incorporated into off-the-shelf search strategies, such as the greedy search and the policy-gradient-based search. Theoretical analyses and extensive experiments on both synthetic and real data demonstrate the efficacy of our method.

## 1 INTRODUCTION

Discovering causal relations among variables is a fundamental problem in various fields such as economics, biology, drug testing, and commercial decision making. Because conducting randomized controlled trials is usually expensive or even infeasible, discovering causal relations from observational data, *i.e.* causal discovery (Pearl, 2000; Spirtes et al., 2000), has received much attention over the past few decades. Early causal discovery algorithms can be roughly categorized into two types: constraint-based ones (*e.g.* PC (Spirtes et al., 2000)) and score-based ones (*e.g.* GES (Chickering, 2002)). In general, these methods cannot uniquely identify the causal graph but are guaranteed to output a Markov equivalence class. Since the seminal work by Shimizu et al. (2006), several methods have been developed, achieving identifiability of the whole causal structure by making use of constrained Functional Causal Models (FCMs), including the linear non-Gaussian model (Shimizu et al., 2006), the nonlinear additive noise model (Hoyer et al., 2009), and the post-nonlinear model (Zhang & Hyvärinen, 2009). Recently, Zheng et al. (2018) proposed a score-based method that formulates the causal discovery problem as continuous optimization with a structural constraint that ensures acyclicity. Based on the continuous structural constraint, several researchers further proposed to model the causal relations by neural networks (NNs) (Lachapelle et al., 2019; Yu et al., 2019; Zheng et al., 2019). Another recent work Zhu & Chen (2019) used reinforcement learning (RL) for causal discovery, where the RL agent searches over the graph space and outputs a graph that fits the data best.

The above approaches are designed for data from a single domain with a fixed causal model, with the limitation that many of the edge directions cannot be determined without strong functional constraints. In addition, the sample size of data from one domain is usually not large enough to guarantee small statistical estimation errors. One way to improve statistical reliability is to combine datasets from multiple domains, such as P-value meta-analyses (Lee, 2015; Marot et al., 2009). The idea of combining multiple-domain data is commonly seen in learning with mixture of Bayesion networks (Thiesson et al., 1998). While mixture of Bayesion networks are usually used for density estimation, the purpose of causal analysis from multiple-domain data is completely different, it aims at discovering the underlying causal graphs for all domains. Regarding causal analysis from multiple-domain data, a challenge is the data heterogeneity problem: the data distribution may vary across domains. For example, in fMRI hippocampus signal analysis, the connection strength among different brain regions may change across different subjects (domains). Due to the distribution shift, directly pooling the data from multiple domains may lead to spurious edges. To tackle the issue, different ways have been investigated, including using sliding windows (Calhoun et al., 2014), online change point detection (Adams & MacKay, 2007), online undirected graph learning (Talih

& Hengartner, 2005), locally stationary structure tracker (Kummerfeld & Danks, 2013), and regime aware learning (Bendtsen, 2016). However, these methods may suffer from high estimation variance due to sample scarcity, large type II errors, and a large number of statistical tests. Huang et al. (2015) recovers causal relations with changing modules by making use of certain types of smoothness of the change, while it does not explicitly locate the changing causal modules. Other similar methods, including Xing et al. (2010); Song et al. (2009), can be reduced to online parameter learning because the causal directions are given.

By utilizing the invariance property (Hoover, 1990; Tian & Pearl, 2001; Peters et al., 2016) and the more general independent change mechanism (Pearl, 2000), recently, Ghassami et al. (2018) developed two methods: identical boundaries (IB) and minimal changes (MC), for causal discovery from multi-domain data. However, the proposed methods 1) assume causal sufficiency (i.e., all common causes of variables are measured), which is usually not held in real circumstances, 2) are designed for linear systems only, 3) and are not capable of identifying causal directions from more than ten domains. Huang et al. (2019) proposed a more general approach called CD-NOD for both linear and nonlinear heterogeneous data, by extending the PC algorithm to tackle the heterogeneity issue. However, inheriting the drawbacks of constraint-based methods, CD-NOD involves a multiple testing problem and is time-consuming due to large number of independence tests.

To overcome the limitations of existing works, we propose a Multiple-Domain Score Search (MDSS) method for causal discovery from heterogeneous data, which enjoys the following properties. (1) To avoid spurious edges when combing multi-domain data, MDSS searches over the space of augmented graphs, which includes an additional domain index as a surrogate variable to characterize the distribution shift. (2) The changing causal modules can be immediately identified from the recovered augmented graph. (3) Benefiting from causal invariance and the independent change mechanism, MDSS uses a novel Multiple-Domain Score (MDS) to help identify more causal directions beyond those in the Markov equivalence class from distribution-shifted data. (4) MDSS can be readily incorporated into off-the-shelf search strategies and is time-efficient and applicable to both linear and nonlinear data. (5) Theoretically, we show that MDSS is guaranteed to find the correct graph skeleton asymptotically, and further identify more causal directions than other traditional score-based and constraint-based algorithms. Empirical studies on both synthetic and real data prove the efficacy of our method.

## 2 METHODOLOGY

In this section, we start from a brief introduction to causal discovery and distribution shifts (Section 2.1), and then in Section 2.2 and 2.3, we introduce our proposed Multiple-Domain Score Search (MDSS). In Section 2.2, MDSS starts with a predefined graph search algorithm to learn the *skeleton* of the causal graph, with the linear Bayesian information criterion (BIC) score or nonlinear generalized score (GS (Huang et al., 2018)) on the augmented causal system. Then in Section 2.3, MDSS further identifies causal *directions* with Multiple-Domain Score (MDS) based on the identified skeleton of the graph from Section 2.2. Both theoretically and empirically, we show that MDSS can identify more directions compared to algorithms that are designed for i.i.d. or stationary data.

### 2.1 BACKGROUND IN CAUSAL DISCOVERY AND DISTRIBUTION SHIFTS

The basic causal discovery problem can be formulated as follows: Suppose there are $d$ observable random variables, *i.e.* $\mathbf{V} = (V_1, ..., V_d)$. Each random variable satisfies the following generating process: $V_i = f_i\left(PA^i, \epsilon\right)$, where $f_i$ is a function to model the causal relation between $V_i$ and its parents $PA^i$, and $\epsilon_i$ is a noise variable with non-zero variance. All the noise variables are independent of each other. The task of *causal discovery* is to recover the causal adjacency matrix $\mathbf{B}$ given the observed data matrix $\mathbf{X} \in \mathbb{R}^{T \times d}$, where $\mathbf{B}_{ij} = 1$ indicates that $V_i$ is a parent of $V_j$, and $T$ is the sample size.

We denote the underlying causal graph over $\mathbf{V}$ as $G_0$. For each $V_i$, we call $P(V_i|PA^i)$ its *causal module*. For a single domain, the joint probability can be factorized as $P(\mathbf{V}) = \prod_{i=1}^{d} P(V_i|PA^i)$. Suppose there are $n$ domains with distribution shifts (*i.e.* $P(\mathbf{V})$ changes across domains), which implies that some causal modules change across domains. The changes may be caused by the variation of functional models, causal strength, or noise variance. Furthermore, we have the following assumptions.

**Assumption 1.** *The changes of causal modules can be represented as functions of domain index $C$, denoted by $g(C)$,*

**Assumption 2.** *There is no confounder in each single dataset, but we allow the changes of different causal modules being dependent.*

**Remark:** If changes in several causal modules are dependent, it can be regarded as special "confounders" that simultaneously affect these causal modules. As a consequence of such confounders, previous causal discovery algorithms designed for i.i.d. or stationary data may output erroneous edges. See section 3.1 for an illustration. Thus, causal discovery from multiple-domain data with distribution shifts (*i.e.* , heterogeneous data) can be much more difficult than that from single-domain data.

## 2.2 SKELETON ESTIMATION ON AUGMENTED GRAPHS

With Assumptions 1 and 2, it is natural to consider $g(C)$ as an extra variable in order to remove any potential influence caused by these special confounders. We assume that there are $L$ such confounders $(g_1(C), ..., g_L(C))$. The causal relation between each observable variable $V_i$ and its parents $PA^i$ can be formalized with

$$V_i = f_i \left( PA^i, \mathbf{g}^i(C), \theta_i(C), \epsilon_i \right), \tag{1}$$

where $\mathbf{g}^i(C) \subseteq \{g_l(C)\}_{l=1}^L$ is the set of confounders that influence $V_i$, $\theta_i(C)$ are the effective parameters in $V_i$'s causal module that are also assumed to be functions of $C$ and are mutually independent for all variables.

Let $G_0$ be the underlying causal graph over $\mathbf{V}$. We denote the graph resulting from adding arrows $\mathbf{g}^i(C) \to V_i$ and $\theta_i(C) \to V_i$ on $G_0$ for each $V_i$ in $\mathbf{V}$ as $G_{aug}$ over $\mathbf{V} \cup \{g_l(C)\}_{l=1}^L \cup \{\theta_i(C)\}_{i=1}^d$. We call $G_{aug}$ an *augmented graph* (see Figure 1(d) as an example), which satisfies the following assumption.

**Assumption 3.** *The joint distribution over* $\mathbf{V} \cup \{g_l(C)\}_{l=1}^L \cup \{\theta_i(C)\}_{i=1}^d$ *is Markov and faithful to* $G_{aug}$.

To remove the potential influence from confounders and recover causal relations from multiple domains, one way is to perform causal discovery algorithms on the augmented graph. While $\{g_l(C)\}_{l=1}^L$ and $\{\theta_i(C)\}_{i=1}^d$ are not directly observed, we take $C$ as a *surrogate variable* (Huang et al., 2019) for them because $C$ is always available as a domain index. Given Assumptions 1, 2 and 3, one can apply any score-based method over $\mathbf{V} \cup \{C\}$ to recover the causal relations among variables $\mathbf{V}$ as if $\{g_l(C)\}_{l=1}^L \cup \{\theta_i(C)\}_{i=1}^d$ were known. For simplicity, we denote the graph over $\mathbf{V} \cup \{C\}$ as augmented graph as well. Since $C$ is the domain index, $P(C)$ follows a discrete uniform distribution. Correspondingly, the generating process of non-stationary data can be considered as follows: First we generate random values from $P(C)$, and then we generate data points over $\mathbf{V}$ according to the SEM in Equation 1. Finally, generated data points are sorted in ascending order according to the values of $C$ (i.e., data points having the same value of $C$ are regarded as belonging to the same domain). In other words, we observe the distribution $P(\mathbf{V}|C)$, where $P(\mathbf{V}|C)$ may change across different values of $C$, resulting in non-stationary data. Note that if we do not include $C$ into the system explicitly, samples of $\mathbf{V}$ are not i.i.d. However, after explicitly including the domain index $C$ into the system, $P(\mathbf{V}, C)$ is fixed, and thus the pooled data are i.i.d. samples from distribution $P(\mathbf{V}, C)$.

Before stating our main result, we first give the definitions of globally consistent scoring criterion and locally consistent scoring criterion, which will be used in the paper.

**Definition 1** (Globally Consistent Scoring Criterion). *Let $D$ be a dataset consisting of $T$ i.i.d. samples from some distribution $P(\cdot)$. Let $\mathcal{H}$ and $\mathcal{G}$ be any DAGs. A scoring criterion $S$ is globally consistent if the following two properties hold as $T \to \infty$:*

    *1. If $\mathcal{H}$ contains $P$ and $\mathcal{G}$ does not contain $P$, then $S(\mathcal{H}, \mathbf{D}) > S(\mathcal{G}, \mathbf{D})$[1].*

    *2. If $\mathcal{H}$ and $\mathcal{G}$ both contain $P$, and $\mathcal{G}$ contains fewer parameters than $\mathcal{H}$, then $S(\mathcal{H}, \mathbf{D}) < S(\mathcal{G}, \mathbf{D})$.*

**Definition 2** (Locally Consistent Scoring Criterion). *Let $D$ be a dataset consisting of $T$ i.i.d. samples from some distribution $P(\cdot)$. Let $\mathcal{G}$ be any DAG, and let $\mathcal{G}'$ be the DAG that results from adding the edge $V_i \to V_j$ on $\mathcal{G}$. A scoring criterion $S(\mathcal{G}, D)$ is locally consistent if the following two properties hold as $T \to \infty$:*

    *1. If $V_j \not\perp\!\!\!\perp V_i | PA_j^{\mathcal{G}}$, then $S(\mathcal{G}', D) > S(\mathcal{G}, D)$.*

    *2. If $V_j \perp\!\!\!\perp V_i | PA_j^{\mathcal{G}}$, then $S(\mathcal{G}', D) < S(\mathcal{G}, D)$.*

---

[1]Here, larger score means the corresponding graph is closer to the equivalent class of the true DAG, while the MDS defined in Section 2.3 should be regarded as a type of "loss function" which needs to be minimized.

It has been shown that the BIC score and the GS score are both globally and locally consistent (Chickering, 2002; Huang et al., 2018).

The procedure for skeleton estimation on augmented graphs is described in Algorithm 1. The predefined graph search algorithms will be discussed in Section 2.4. Apart from the recovered skeleton over $\mathbf{V}$, the changing modules can be detected as well in Step 4 of Algorithm 1. It is important to note that we allow causal relations to be either linear or nonlinear. If they are nonlinear, we apply GS as a score function. When they are linear, although we can also use GS, we use linear BIC instead because it is less likely to be overfitting for linear data and is computationally more efficient.

---

**Algorithm 1** Skeleton Search on Augmented Graph

    **Input:** $n$ datasets, each has $T$ observations, $d$ variables and index $C$.
    **Output:** skeleton $\mathcal{S}$ of $G_{aug}$'s subgraph $G_1$ over $\mathbf{V}$, and variables $\mathbf{V}_C \in \mathbf{V}$ that are connected with $C$.
1: Pool all datasets with an extra surrogate variable $C$ to form a data matrix $\mathbf{X} \in \mathbb{R}^{nT \times (d+1)}$.
2: Use the predefined graph search algorithm with BIC or GS plus acyclicity constraints to recover the augmented graph. Eliminate any direction $V_i \to C$ in the graph with the prior that any variable $V_i$ does not affect domain index. This step leads to the recovered augmented graph $G_{aug}$.
3: Discard the index variable in $G_{aug}$ to obtain the induced subgraph $G_1$. Discard the directions in $G_1$ and output the skeleton $\mathcal{S}$ of $G_1$.
4: Detect changing causal modules by inspecting $G_{aug}$ recovered in Step 2, and output $\mathbf{V}_C$.

---

The validity of searching on augmented graph is guaranteed by Theorem 1.

**Theorem 1.** *Let $D$ be the pooling of all datasets, $D_C$ be the augmented dataset with the domain index as an extra random variable. Let $G_0$ be the underlying causal graph for the distribution of $D$ over $\mathbf{V}$, $G_C$ be the underlying causal graph for the distribution of $D_C$ over $\mathbf{V} \cup C$. If we denote $G'_C$ as the graph after the following modifications on $G_C$: 1. adding any edges, 2. deleting any edges or 3. reversing any edges that changes the conditional dependence relation of $G_C$, then we have $S(G_C, D_C) > S(G'_C, D_C)$, where $S$ is any globally consistent scoring criterion.*

Proof of the theorem is given in Appendix A.1. Intuitively, this theorem means we will obtain an augmented graph that is in the same Markov equivalence class as the true augmented graph if we maximize the score.

## 2.3 CAUSAL DIRECTION DETERMINATION BY MULTIPLE-DOMAIN-SCORE

For each variable $V_k \in \mathbf{V}_C$, we prove that it is possible to determine the directions of edges that connect to $V_k$. We denote $V_l$ as any variable that connects to $V_k$. There are two possible cases:

**1)** $V_l \notin \mathbf{V}_C$. In this case, $C - V_k - V_l$ forms an unshielded triple. It is intuitive to incorporate the prior that $C \to V_k$ (*i.e.* change of domain leads to the distribution shift of $V_k$). There are two possible patterns in this case: $C \to V_k \to V_l$ and $C \to V_k \leftarrow V_l$, which we denote as $\mathcal{P}$ and $\mathcal{P}'$ respectively. For $\mathcal{P}$, the causal mechanisms $P(\text{effect}|\text{cause})$ is invariant when $P(\text{cause})$ changes. For $\mathcal{P}'$, we have the invariance of $P(\text{cause})$ when the causal mechanism $P(\text{effect}|\text{cause})$ changes, which is complementary to the invariance of causal mechanisms. The direction between $V_k$ and $V_l$ can be determined as long as a globally consistent score is used. To be specific, suppose $\mathcal{P}$ is the true causal pattern underlying the generative distribution, the score of $\mathcal{P}$ will be larger than that of $\mathcal{P}'$ if the score function used is globally consistent and decomposable, because compared with $\mathcal{P}$, $\mathcal{P}'$ eliminates a conditional independence ($C \perp\!\!\!\perp V_l | V_k$) that actually holds in the generative distribution. This causal direction determination is not achievable for algorithms designed for stationary data from a single domain (because domain index cannot be used as an additional variable in this case). To utilize this prior, we simply eliminate any direction $V_i \to C$ as described in Step 2 of Algorithm 1. Figure 1(a) is a graphical illustration for this case.

**2)** $V_l \in \mathbf{V}_C$. In this case, both $V_k$ and $V_l$ are connected to $C$, which is much more difficult than case 1). We propose a novel multiple-domain score (MDS) that utilizes the property of independent changes of causal modules to determine causal directions based on the causal skeleton derived from Algorithm 1. To specify the idea, we take the two-variable case as an example. Here we assume the true causal direction is $V_1 \to V_2$. Figure 1(b) stands for the case where $\theta_1$ and $\theta_2$ are independent. (We drop the notation of domain index $C$ for simplicity). In other words, $P(V_1; \theta_1)$ and $P(V_2|V_1; \theta_2)$ change independently. If the recovered

direction is reverse (see Figure 1(c)), we factorize the joint distribution as

$$P(V_1, V_2; \theta_1', \theta_2') = P(V_2; \theta_2') P(V_1|V_2; \theta_1'), \quad (2)$$

where $\theta_1'$ and $\theta_2'$ are assumed to be sufficient for $P(V_2)$ and $P(V_1|V_2)$ respectively. Since $V_1 \leftarrow V_2$ is not the true direction, $\theta_1'$ and $\theta_2'$ are not independent, and they are determined jointly by $\theta_1$ and $\theta_2$. Based on this point, the causal direction can be determined by comparing the dependence between $\theta_1$, $\theta_2$ and the dependence between $\theta_1'$ and $\theta_2'$, and choose the direction with smaller dependence.

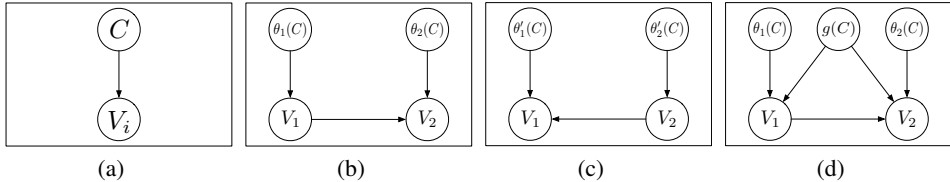

(a)  (b)  (c)  (d)

Figure 1: (a) In skeleton estimation on augmented graph, we force the direction to be $C{\rightarrow}V_i$ if algorithm finds a link between them. (b)(c) True causal graph for $V_1$ and $V_2$ and the graph with reversed causal direction in the two-variable example. (c) A two-variable example where confounder exists.

For linear systems, the dependence can be described with covariance. $\theta_1$ and $\theta_2$ can be easily obtained by regressing $V_1$ and $V_2$ on their parents respectively. We first perform the regression for each domain then calculate the covariance between $\theta_1(C)$ and $\theta_2(C)$. When there are more than two variables that are connected to $C$, we denote such set of variables as $\mathbf{V}_C$ with cardinality $m$. For each variable $V_C^k \in \mathbf{V}_C$ and its parents $PA_C^k \subseteq \mathbf{V}_C$, we calculate the sum of the dependence between parameters of $V_C^k$'s causal module and the parameters of the causal module of each variable in $PA_C^k$. To incorporate the minimization of such dependence into score-based method, we propose MDS for linear systems:

$$MDS_{linear} = \frac{1}{n} \sum_{i=1}^{n} (d\ln(T_i) - 2\ln(L_i)) + \lambda_1 \mathbf{I}(\mathcal{G} \notin \text{DAGs}) + \lambda_2 h(A) + \frac{\lambda_3}{m} \sum_{k=1}^{m} |cov(\theta_{V_C^k}, \theta_{PA_C^k})|, \quad (3)$$

where $n, d, T_i$ and $L_i$ represent the number of domains, the number of variables (here we assume this quantity is the same for all domains), sample size, and the maximized log likelihood for domain $i$, respectively. $\lambda_1$, $\lambda_2$ and $\lambda_3$ are regularization coefficients. $\lambda_3$ is fixed to 0.001 in our experiments, $\lambda_1$ and $\lambda_2$ are adjusted dynamically while training (see Zhu & Chen (2019) for how $\lambda_1$ and $\lambda_2$ are adjusted). $A$ is the weighted adjacency matrix recovered by the algorithm. $m$ is the number of nonstationary variables. $cov(\cdot)$ is the covariance operator. The first term in Equation 3 is the average of BIC on $n$ domains, the second and third terms are acyclicity constraints proposed in Zhu & Chen (2019) to narrow down the search space to DAGs. See Appendix A.2 for more details about the acyclicity constraints.

For nonlinear systems, $\theta$ cannot be calculated explicitly. In the two-variable case, the dependence between $\theta_1$ and $\theta_2$ can be characterized by the dependence between $P(V_1)$ and $P(V_2|V_1)$ with the assumption that $\theta_1$ and $\theta_2$ are sufficient for the corresponding distribution module. To calculate the dependence between $P(V_1)$ and $P(V_2|V_1)$, Huang et al. (2019) proposes to first use kernel embeddings of distributions $P(V_1)$ and $P(V_2|V_1)$, then measure their dependence with extended Hilbert Schmidt Independence Criterion (HSIC (Gretton et al., 2008)) in Reproducing Kernel Hilbert Space (RKHS). When there are more than two variables that are connected to $C$, for each variable $V_C^k \in \mathbf{V}_C$ and its parents $PA_C^k \in \mathbf{V_C}$, we calculate the dependence between $P(PA_C^k)$ and $P(V_C^k|PA_C^k)$. We propose corresponding MDS for nonlinear systems by integrating such dependence with GS:

$$MDS_{nonlinear} = \frac{1}{n} \sum_{i=1}^{n} (GS_i) + \lambda_1 \mathbf{I}(\mathcal{G} \notin \text{DAGs}) + \lambda_2 h(A) + \frac{\lambda_3}{m} \sum_{k=1}^{m} HSIC(\mu_{V_C^k|PA_C^k}, \mu_{PA_C^k}), \quad (4)$$

where $GS_i$ is the generalized score for domain $i$, $\mu_{V_C^k|PA_C^k}$ and $\mu_{PA_C^k}$ are the kernel embeddings of distributions $P(V_C^k|PA_C^k)$ and $P(PA_C^k)$ respectively. $HSIC(\cdot)$ is HSIC operator that measures the dependence of two random variables. See Appendix A.3, A.4 and A.5 for brief descriptions of GS, kernel embedding of distributions and HSIC respectively.

**Degeneration issue.** If we apply search strategies over the entire space of graphs over $\mathbf{V}$ to optimize MDS, the MDS penalty (corresponds to the fourth term in Equation 3 and 4) tends to eliminate any edges between each $V_C^k$ and its parents because the dependence between $\theta_{V_C^k}$ and empty set (*i.e.* no parents for $V_C^k$) is 0. We call this an degeneration issue. To tackle this issue, we optimize the MDS score based on the skeleton of graph $G_1$ from Algorithm 1. To be specific, we fix the skeleton $\mathcal{S}$ of $G_1$ and apply search strategies over the space defined by $\mathcal{S}$ to optimize the MDS score. In other words, MDS is optimized by only altering the direction of each edge in $\mathcal{S}$. With the solution to degeneration issue, we claim that the proposed MDS can recover more correct directions compared with $G_1$. This is supported by Theorem 2 and Theorem 3. See Appendix A.6 and A.7 for proofs.

Let $\mathbf{D}$ be the pooling of $n$ datasets with distribution shifts and $G_0$ be the DAG underlying the distribution of $\mathbf{D}$. Let $\mathcal{S}$ be the skeleton of $G_0$ and $G_1$ (recall $G_1$ is in the same equivalent class as $G_0$, which means they have the same skeleton). Let $\mathbf{E_1}$ be the set of edges that exist in both $G_0$ and $G_1$ but have different directions in $G_0$ and $G_1$. Let $\mathbf{E_2} \in \mathbf{E_1}$ be the set of edges whose left node (or variable) and right node are both nonstationary. Let $n_1$ and $n_2$ be the cardinality of $\mathbf{E_1}$ and $\mathbf{E_2}$.

**Theorem 2.** *For linear systems, let $G^* = \underset{G}{\arg\min}\ MDS_{linear}(G, \mathbf{D})$, let $\mathbf{E_2^*} \in \mathbf{E_2}$ be the set of edges whose directions are correctly determined by $G^*$, let $n^*$ denote the cardinality of $\mathbf{E_2^*}$. Given $\mathbf{E_2}$ is not empty and $G_1$, $G^*$ have the same skeleton $\mathcal{S}$, then $G^*$ is in the same equivalent class as $G_0$ and $n^* = n_2$.*

**Theorem 3.** *For nonlinear systems, let $G^* = \underset{G}{\arg\min}\ MDS_{nonlinear}(G, \mathbf{D})$, let $\mathbf{E_2^*} \in \mathbf{E_2}$ be the set of edges whose directions are correctly determined by $G^*$, let $n^*$ denote the cardinality of $\mathbf{E_2^*}$. Given $\mathbf{E_2}$ is not empty and $G_1$, $G^*$ have the same skeleton $\mathcal{S}$, then $G^*$ is in the same equivalent class as $G_0$ and $0 \leqslant n^* \leqslant n_2$.*

Theorem 2 and 3 mainly state that under proper assumptions, the directions of some edges whose left node and right are both nonstationary can be correctly determined by the proposed method.

**Confounding case.** When the confounder $g(C)$ exists (*e.g.* Figure 1(d)), the above approach still works if the influence from the confounder is not very strong for the following reason: for the correct direction, the dependence of $\theta_1$ and $\theta_2$ would come from the confounder, while for the wrong direction, the dependence would come from the confounder as well as the wrong direction.

## 2.4 GRAPH SEARCH STRATEGIES

The proposed MDSS can be readily incorporated into off-the-shelf search strategies. In this paper, we adopt the policy-gradient-based search strategy (Zhu & Chen, 2019) to search for the optimal causal structure. Compared with other search strategies such as greedy equivalence search (GES (Chickering, 2002)), max-min hill climbing (Tsamardinos et al., 2006), direct search by regarding the weighted graph adjacency matrix as parameters (Zheng et al., 2018; Yu et al., 2019; Lachapelle et al., 2019), the policy-gradient-based search by a reinforcement learning (RL) agent with stochastic policy can determine automatically where to search given the uncertainty information of the learned policy, which gets updated promptly by the stream of reward signals (Zhu & Chen, 2019). The graph search strategy using RL is proven to be better than other search strategies mentioned above empirically.

The idea of causal discovery with RL can be summarized as follows. The algorithm uses an encoder-decoder neural network model to generate directed graphs from the observed data, which are then used to compute rewards consisting of the predefined score function as well as some regularization terms for acyclicity. The encoder-decoder model can be regarded as an "actor" that learns to generate "actions" (i.e., graph adjacency matrices) in actor-critic algorithm, an algorithm commonly used in RL. The reward function can be regarded as the "environment" that evaluates how good the "action" is (i.e., how good the produced graph adjacency matrix fits the observed data). The weights of the encoder-decoder model is trained by policy gradient and stochastic optimization methods. The output of the algorithm is the graph that achieves the best reward during the training process.

To integrate MDS with the policy-gradient-based search, we replace the predefined score function in the original paper (where BIC is used) with MDS. Apart from policy-gradient-based search, we also experiment with greedy equivalence search, details of which can be found in Appendix A.8.

The complete search procedure is described in Algorithm 2.

---

**Algorithm 2** Multiple-domain Score Search

    **Input** $n$ datasets each has $T$ observations, $d$ variables and index $C$.
    **Output** causal graph $G_2$ over $\mathbf{V}$.

  1: Execute Algorithm 1, input all the datasets and corresponding domain index, output skeleton $\mathcal{S}$ and nonstationary variables $\mathbf{V}_C$.
  2: Execute the predefined graph search algorithm with MDS in the space defined by $\mathcal{S}$, output $G_2$ over $\mathbf{V}$.
  3: Perform any pruning methods on $G_2$ if needed.

---

## 3 EXPERIMENTS

In this section, we conduct empirical studies to show the effectiveness of our MDSS method combined with the MDS score. We compare MDSS to some well-known causal discovery algorithms that are designed for i.i.d. or stationary data from a single domain (GES (Chickering, 2002), PC (Spirtes et al., 2000), LiNGAM (Shimizu et al., 2006), NO-TEARS (Zheng et al., 2018) and RL (Zhu & Chen, 2019)) as well as algorithms designed for heterogeneous data from multiple domains (CD-NOD (Huang et al., 2019), MC and IB (Ghassami et al., 2018)). The comparison is made on both synthetic and real data.

We evaluate the estimated graphs using three metrics: True Negative Rate (TNR), True Positive Rate (TPR), and Structural Hamming Distance (SHD, *i.e.* , the smallest number of edge additions, deletions, and reversals to convert the estimated graph into the true DAG). A lower SHD indicates a better estimate of the causal graph. For algorithms that output completed partially directed acyclic graph (CPDAG), we randomly choose a direction for those undirected edges.

### 3.1 A TOY EXAMPLE

We use a synthetic toy example to illustrate the influence of confounders $g(C)$ for algorithms (we use RL) designed for homogeneous data, and demonstrate that MDSS can avoid such influence and further identify more directions. See Appendix A.9 for this example.

### 3.2 SYNTHETIC DATA

In this section, we conduct extensive experiments with MDSS and other causal discovery algorithms on linear and nonlinear synthetic data. We denote $n$ as the number of datasets, each has $d$ variables and $T$ observations. We set $n \in \{6, 7, 8, 9, 10, 11, 12, 13, 14, 15, 16, 17, 18, 19, 20\}$, $d \in \{6, 7, 8\}$, $T = 100$ for both linear and nonlinear data. We repeat each setting 20 times with DAGs randomly generated by Erdős–Rényi model (ER) with parameter 0.3. Each variable $V_i$ is chosen as nonstationary with probability 0.6. Similar to Section 3.1, linear data are generated using linear SEM $V_i = w_i P A_i + b_i + \epsilon_i$, we fix $w_i$ and $\epsilon_i$ across domains and vary $b_i$ if $V_i$ is chosen as nonstationary. Nonlinear data is generated using nonlinear SEM $V_i = f_i(PA_i) + b_i + \epsilon_i$, $f_i(\cdot)$ is randomly picked from $\{sin(\cdot), cos(\cdot), sigmoid(\cdot)\}$, and $b_i$ varies if $V_i$ is nonstationary, $\epsilon_i$ stays invariant.

We first consider the setting when $n = 6$ and $d = 10$. MDSS, MC, IB, and CD-NOD are tested on data from all domains. GES, PC, LiNGAM, NO-TEARS, and RL are tested on data from all domains as well as data from a single domain (the domain is randomly chosen). For GES, we use fast GES (FGES (Ramsey et al., 2017)), which is an improved version of the original GES. The mean and standard deviation are reported in Table 1 and 2. As we can see, MDSS outperforms other algorithms on both linear and nonlinear data. The performances of PC, FGES, LiNGAM, NO-TEARS, and RL on pooling of all domain data are worse than that on single domain data. Specifically, despite the minor increase in TPR, their TNR decrease dramatically when data from all domains are used. This phenomenon further proves our proposition that distribution shifts will introduce spurious edges if not properly dealt with.

We further compare the performance of MDSS, IB, MC and CD-NOD by varying $d$ and $n$. The results are reported in Figure 2. The black curves in figures of row 2 and row 3 are shorter than others because CD-NOD takes too much time to give any result when $d > 6$ and $n > 15$. According to the results, MDSS outperforms the others in most cases.

To demonstrate that the proposed MDS contributes to the performance, we conduct some ablation studies. To be specific, we keep the directions in step 3 of Algorithm 1 and output $G_1$ (Algorithm 2, or MDS search, is not executed). We use the same experimental setting as Table 1. The results (TPR, TNR and SHD) are

Table 1: Empirical results for MDSS, MC, IB and CD-NOD on linear and nonlinear data.

|  |  | MDSS | MC | IB | CD-NOD |
|---|---|---|---|---|---|
| Linear | TPR | 0.98±0.08 | 0.51±0.37 | 0.40±0.20 | 0.60±0.25 |
|  | TNR | 0.90±0.11 | 0.65±0.20 | 0.55±0.17 | 0.92±0.09 |
|  | SHD | 1.25±1.26 | 4.35±2.35 | 5.45±1.56 | 1.70±1.19 |
| Nonlinear | TPR | 0.65±0.17 | 0.18±0.18 | 0.23±0.19 | 0.65±0.30 |
|  | TNR | 0.80±0.15 | 0.78±0.11 | 0.82±0.09 | 0.64±0.14 |
|  | SHD | 2.30±1.39 | 5.20±1.25 | 4.65±1.31 | 2.60±1.20 |

Table 2: Empirical results for PC, FGES, LiNGAM, NO-TEARS and RL on linear and nonlinear data.

|  |  | PC | | FGES | | LiNGAM | | NO-TEARS | | RL | |
|---|---|---|---|---|---|---|---|---|---|---|---|
|  |  | single | pool | single | pool | single | pool | single | pool | single | pool |
| Linear | TPR | 0.65±0.25 | 0.78±0.21 | 0.70±0.29 | 0.85±0.17 | 0.23±0.21 | 0.20±0.22 | 0.68±0.18 | 0.73±0.13 | 0.81±0.11 | 0.87±0.10 |
|  | TNR | 0.48±0.20 | 0.28±0.42 | 0.44±0.32 | 0.47±0.22 | 0.69±0.06 | 0.35±0.22 | 0.71±0.10 | 0.60±0.13 | 0.87±0.09 | 0.37±0.22 |
|  | SHD | 6.00±2.00 | 6.60±2.69 | 5.60±2.73 | 5.10±1.51 | 3.40±0.66 | 7.40±2.42 | 4.20±1.21 | 4.95va1.45 | 1.50±1.16 | 7.50±1.24 |
| Nonlinear | TPR | 0.35±0.28 | 0.53±0.31 | 0.35±0.32 | 0.48±0.28 | 0.13±0.13 | 0.28±0.24 | 0.51±0.37 | 0.55±0.32 | 0.52±0.28 | 0.25±0.18 |
|  | TNR | 0.43±0.28 | 0.18±0.56 | 0.44±0.26 | 0.22±0.45 | 0.75±0.13 | 0.48±0.24 | 0.85±0.11 | 0.69±0.10 | 0.75±0.10 | 0.67±0.14 |
|  | SHD | 7.30±2.45 | 7.80±3.25 | 7.30±3.10 | 8.10±3.27 | 4.30±0.9 | 7.50±2.58 | 2.55±1.48 | 3.05±1.14 | 4.10±1.22 | 5.00±1.09 |

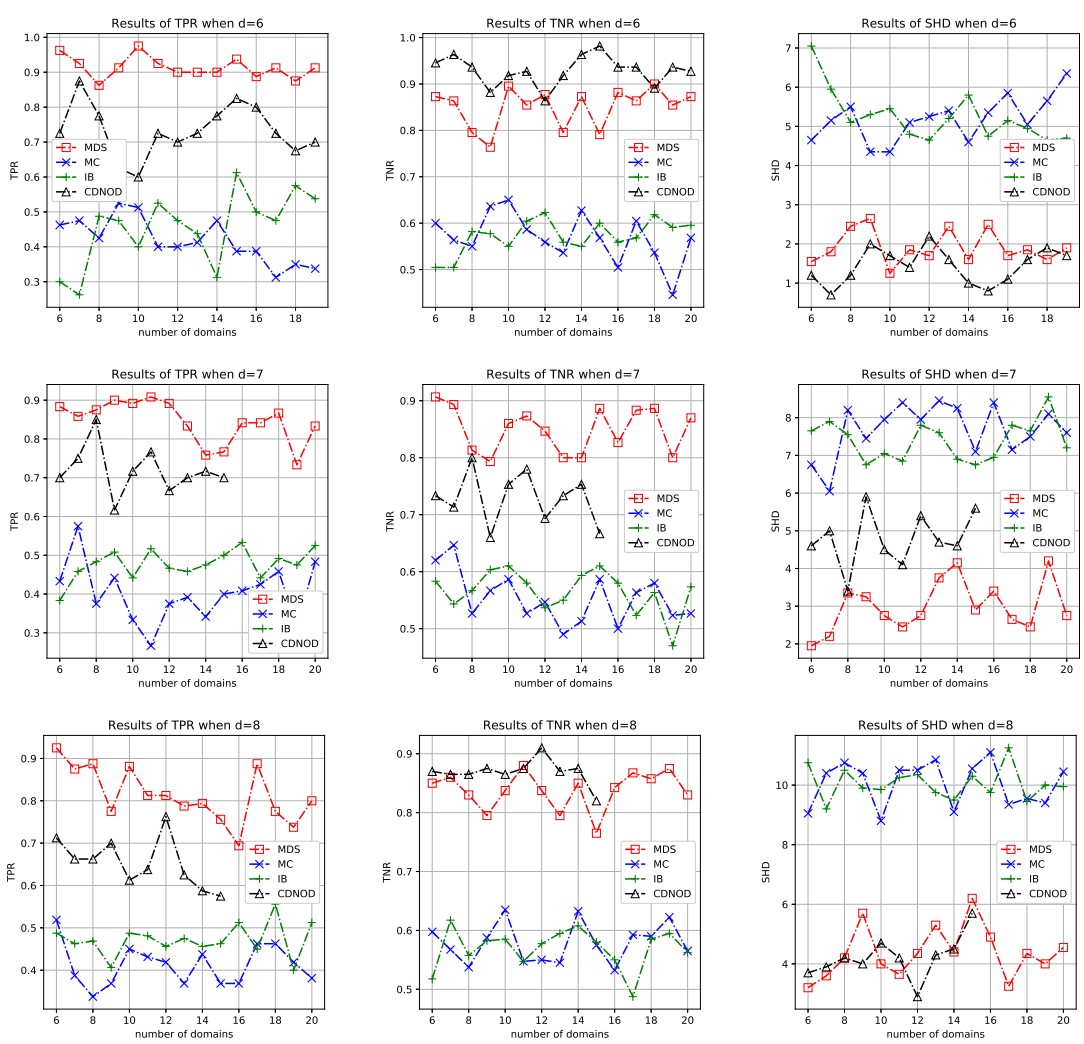

Figure 2: Comparison of MDSS, MC, IB and CD-NOD when $n$ and $d$ vary.

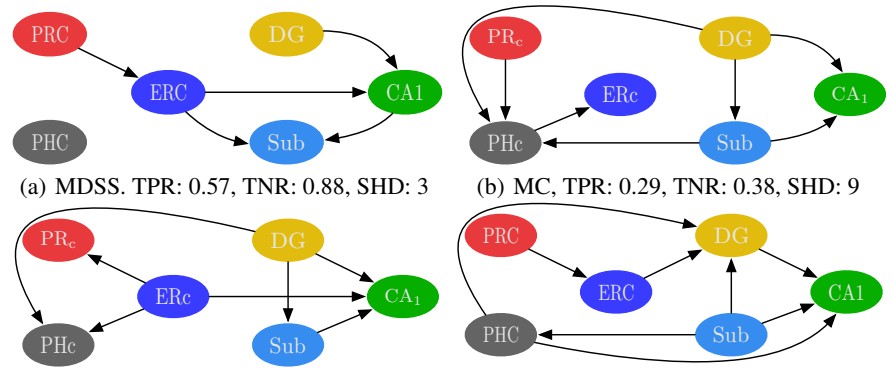

(a) MDSS. TPR: 0.57, TNR: 0.88, SHD: 3  (b) MC, TPR: 0.29, TNR: 0.38, SHD: 9

(c) IB. TPR: 0.29, TNR: 0.38, SHD: 7  (d) CD-NOD. TPR: 0.43, TNR: 0.38, SHD: 8

Figure 3: Recovered graphs from MDSS, MC, IB and CD-NOD on hippocampus data.

$0.58 \pm 0.21$, $0.92 \pm 0.07$ and $7.95 \pm 4.51$ for linear data, $0.32 \pm 0.22$, $0.58 \pm 0.22$ and $6.20 \pm 0.26$ for nonlinear data. When compared with the results of MDSS in Table 1, it is obvious that MDS search identifies more directions and boost the performance.

## 3.3 REAL DATA

We apply MDSS to fMRI hippocampus dataset (Poldrack et al., 2015). This dataset records signals from 6 separate brain regions: perirhinal cortex (PRC), parahippocampal cortex (PHC), entorhinal cortex (ERC), subiculum (Sub), CA1, and CA3/Dentate Gyrus (DG) of a single person with resting states in 84 successive days. The records for each day can be regarded as a domain. We select 10 of them. The results from MDSS, MC, IB and CD-NOD are given in Figure 3. The directions of anatomical ground truth are: PHC → ERC, PRC → ERC, ERC → CA3/DG, CA3/DG → CA1, CA1 → Sub, Sub → ERC and ERC → CA1. See Appendix A.10 for illustration of the ground true causal graph. As we can see, MDSS outperforms the others.

## 4 CONCLUSIONS

This paper proposes a Multiple-Domain Score Search (MDSS) algorithm for causal discovery from heterogeneous data. It first performs skeleton learning over the space of augmented graphs. Then a Multiple-Domain Score (MDS) is used to determine causal directions based on the skeleton of the recovered graph. The MDS is proposed based on distribution shifts across domains and the assumption of independent change. Compared with previous methods, MDSS can remove the influence of distribution shifts and further recover more causal directions. In future work, we aim to improve MDSS from the following two aspects: 1) Score calculation takes more time than training NNs for searching, so it is essential to optimize score computing to accelerate the entire framework. 2) The current framework of MDSS cannot deal with the more general case where causal directions also change, while this phenomenon does exist in some real-world circumstances.

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

## A  APPENDIX

### A.1  PROOF OF THEOREM 1

*Proof.* Let $P(\mathbf{V})$ be the distribution of the data over $V$, $P_C(\mathbf{V}, C)$ be the distribution of the augmented data over $\mathbf{V} \cup C$. According to Markov and faithfulness conditions, $G_C$ is the perfect map of $P_C(\mathbf{V}, C)$.

1. If $G'_C$ is the graph after deleting any edges from $G_C$, then $G_C$ contains $P_C(\mathbf{V}, C)$ while $G'_C$ does not. According to Definition 1, we have $S(G_C, D_C) > S(G'_C, D_C)$.

2. If $G'_C$ is the graph after reversing any edges from $G_C$, that changes the conditional dependence of $G_C$, then obviously $G_C$ contains $P_C(\mathbf{V}, C)$ while $G'_C$ does not. According to Definition 1, we have $S(G_C, D_C) > S(G'_C, D_C)$.

3. If $G'_C$ is the graph after adding any edges from $G_C$. Although both $G'_C$ and $G_C$ contain $P_C(\mathbf{V}, C)$, $G'_C$ has more parameters. According to Definition 1, we have $S(G_C, D_C) > S(G'_C, D_C)$. ☐

## A.2  ACYCLICITY CONSTRAINTS

Causal discovery from samples of a joint distribution is a challenging combinatorial problem because of the intractable search space super-exponential in the number of graph nodes. Recently, Zheng et al. (2018) formulates the structure learning problem as a purely continuous optimization problem by a new characterization of acyclicity that is not only smooth but also exact. Zheng et al. (2018) proposes to measure the "DAG-ness" of a graph by

$$h(A) = \operatorname{tr}\left(e^{A \circ A}\right) - d, \tag{5}$$

where $A$ is a weighted adjacency matrix and $d$ is the number of node in the graph. Function $h(\cdot)$ satisfies the following properties:

- $h(A) = 0$ if and only if $A$ is acyclic.
- The values of h quantify the "DAG-ness" of the graph.
- $h$ is smooth.
- $h$ and its derivatives are easy to compute.

Further, Zhu & Chen (2019) finds that $h(A)$, which is non-negative, can be small for certain cyclic graphs and its minimum over all non-DAGs is not easy to compute. As a consequence, it would require a very large penalty weight for $h(A)$ to obtain exact DAGs if only $h(A)$ is used. To address the issue, Zhu & Chen (2019) proposes another acyclicity penalty term $\mathbf{I}(\mathcal{G} \notin \mathrm{DAGs})$, which is the indicator function w.r.t. acyclicity to induce exact DAGs. The combination of the above two acyclicity constraints can be written as $\lambda_1 \mathbf{I}(\mathcal{G} \notin \mathrm{DAGs}) + \lambda_2 h(A)$, which corresponds to the second and third terms in our proposed MDS.

See their original papers Zheng et al. (2018) and Zhu & Chen (2019) for more details.

## A.3  GENERALIZED SCORE

We use generalized score (GS (Huang et al., 2018)) as a model selection criteria to measure how well the a graph fits the data. Here we give a brief introduction of the calculation of GS.

Assume $X$ is a random variable with domain $\mathcal{X}$, and $\mathcal{H}_{\mathcal{X}}$ is a reproducing kernel Hilbert space (RKHS) on $\mathcal{X}$ with continuous feature mapping $\phi_{\mathcal{X}} : \mathcal{X} \to \mathcal{H}_{\mathcal{X}}$. Similarly we define variable $Y$, $Z$ with domain $\mathcal{Y}$, $\mathcal{Z}$, the corresponding RKHS $\mathcal{H}_{\mathcal{Y}}$, $\mathcal{H}_{\mathcal{Z}}$ and feature mapping $\phi_{\mathcal{Y}}$, $\phi_{\mathcal{Z}}$. Let $\ddot{Z} := (Y, Z)$, consider the following two regression functions in RKHS:

$$\begin{aligned} \phi_{\mathcal{X}}(X) &= F_1(Z) + U_1, \\ \phi_{\mathcal{X}}(X) &= F_2(\ddot{Z}) + U_2, \end{aligned} \tag{6}$$

where $F_1 : \mathcal{Z} \to \mathcal{H}_{\mathcal{X}}$ and $F_2 : \ddot{\mathcal{Z}} \to \mathcal{H}_{\mathcal{X}}$. If $X \perp\!\!\!\perp Y | Z$, the following equation holds:

$$E_Z[Var_{X|Z}[\phi_{\mathcal{X}}(X)|Z]] = E_{\ddot{Z}}[Var_{X|\ddot{Z}}[\phi_{\mathcal{X}}(X)|\ddot{Z}]], \tag{7}$$

which means that it is not useful to incorporate $Y$ as a predictor of $X$ given $Z$, so the first model (*i.e.* the model with less complexity) in Equation 6 is preferred.

Cross-validated likelihood is used to express such preference. To perform cross validation, the whole data set $D$ is split into a training set and a test set and repeat this procedure $K$ times, *i.e.* $K$-fold cross validation. Let $D_1^{(k)}$ and $D_0^{(k)}$ be the kth training set and kth validation set respectively. Let $D_{1,i}^{(k)}$ and $D_{0,i}^{(k)}$ be the data of $X_i$ and its parents in training set and validation set respectively. The GS of DAG $\mathcal{G}_h$ using cross-validated likelihood is calculated with

$$\begin{aligned} S_{\mathrm{CV}}\left(\mathcal{G}_h; D\right) &= \sum_{i=1}^{m} S_{\mathrm{CV}}\left(X_i, PA_i^{\mathcal{G}_h}\right) \\ &= \sum_{i=1}^{m}\left(\frac{1}{K}\sum_{k=1}^{K} \ell\left(\hat{\tilde{F}}_i^{(k)}|D_{0,i}^{(k)}\right)\right), \end{aligned} \tag{8}$$

where $PA_i^{\mathcal{G}_h}$ are parents of $X_i$ in graph $\mathcal{G}_h$, $\hat{\tilde{F}}_i^{(k)}$ is the regression function estimated from kth training data $D_{1,i}^{(k)}$, $\ell\left(\hat{\tilde{F}}_i^{(k)}|D_{0,i}^{(k)}\right)$ is the log-likelihood evaluated on the kth validation set with the estimated regression function.

Another type of GS based on marginal likelihood are also proposed, see Huang et al. (2018) for more details.

### A.4 KERNEL EMBEDDING OF DISTRIBUTIONS

According to Equation 4 in our paper, we need to calculate the dependence between distributions $P(V_C^k|PA^k)$ and $P(PA^k)$. The dependence between random variables can be measured by Hilbert Schmidt Independence Criterion (HSIC), which will be discussed in next section. To transform the distribution for data from different domains to a random variable in RKHS, we use kernel embeddings of conditional distributions (Song et al., 2013). In the rest of this section, we denote $PA^k$ as $X$ and $V_C^k$ as $Y$ for simplicity.

Let $\mathcal{X}$ be the domain of $X$, and $(\mathcal{H}, k)$ be a reproducing kernel Hilbert space (RKHS) with a measurable kernel on $\mathcal{X}$. Let $\phi(x) \in \mathcal{H}$ be a continuous feature mapping $\phi_{\mathcal{X}} : \mathcal{X} \rightarrow \mathcal{H}$. Similar notations are for variables $Y$ and $C$. We define the cross-covariance operator $C_{YX} : \mathcal{H} \rightarrow \mathcal{G}$ as $C_{YX} := E_{YX}[\phi(X) \otimes \psi(Y)]$.

The kernel embedding of the conditional distribution $P(X|C=c_n)$ for data from a given domain $C=c_n$ can be calculated as

$$\mu_{X|C=c_n} = \mathcal{C}_{XC}\mathcal{C}_{CC}^{-1}\phi(c_n) \tag{9}$$

The empirical estimate for $\mu_{X|C=c_n}$ is

$$\hat{\mu}_{X|C=c_n} = \frac{1}{N}\mathbf{\Phi_x}\mathbf{\Phi_c}^{\top}\left(\frac{1}{N}\mathbf{\Phi_c}\mathbf{\Phi_c}^{\top} + \lambda I\right)^{-1}\phi_{c_n}, \tag{10}$$
$$= \mathbf{\Phi_x}\left(\mathbf{K_c} + \lambda I\right)^{-1}\mathbf{k}_{\mathbf{c},c_n}$$

where $N$ is sample size, $\mathbf{\Phi_x} := [\phi(x_1), \dots, \phi(x_N)]$, $\mathbf{\Phi_c} := [\phi(c_1), \dots, \phi(c_N)]$, $\mathbf{K_c}(c_t, c_{t'}) = \langle \phi(c_t), \phi(c_{t'}) \rangle$, $\mathbf{k}_{\mathbf{c},c_n} := [k(c_1, c_n), \dots, k(c_N, c_n)]^{\top}$. The corresponding Gram matrix with Gaussian kernel with $\sigma_x$ is

$$\mathbf{M}_X^{\mathcal{H}} = \exp\left(-\frac{\operatorname{diag}\left(\mathbf{M}_X^l\right) \cdot \mathbf{1}_N + \mathbf{1}_N \cdot \operatorname{diag}\left(\mathbf{M}_X^l\right) - 2\mathbf{M}_X^l}{2\sigma_x^2}\right), \tag{11}$$

where $\operatorname{diag}(\cdot)$ sets all off-diagonal entries of the matrix as zero, and $\mathbf{1}_N$ is a $N \times N$ matrix with all entries being 1. $\mathbf{M}_X^l$ is the Gram matrix with a linear kernel:

$$\mathbf{M}_X^l = \mathbf{K_c}\left(\mathbf{K_c} + \lambda I\right)^{-1}\mathbf{K_x}\left(\mathbf{K_c} + \lambda I\right)^{-1}\mathbf{K_c}, \tag{12}$$

whose $(c, c')$ entry can be calculated by

$$\begin{aligned}\mathbf{M}_X^l(c, c') &= \hat{\mu}_{X|C=c}^{\top}\hat{\mu}_{X|C=c'} \\ &= \mathbf{k}_{\mathbf{c},c}^{\top}\left(\mathbf{K_c} + \lambda I\right)^{-1}\mathbf{\Phi_x}^{\top}\mathbf{\Phi_x}\left(\mathbf{K_c} + \lambda I\right)^{-1}\mathbf{k}_{\mathbf{c},c'} \\ &= \mathbf{k}_{\mathbf{c},c}^{\top}\left(\mathbf{K_c} + \lambda I\right)^{-1}\mathbf{K_x}\left(\mathbf{K_c} + \lambda I\right)^{-1}\mathbf{k}_{\mathbf{c},c'}\end{aligned} \tag{13}$$

Similarly we can calculate the empirical kernel embedding of the conditional distribution $P(Y|X, C=c_n)$ and the corresponding Gram matrix, which we denote as $\hat{\mu}_{Y|X,C=c_n}$ and $\mathbf{M}_{Y|X}^{\mathcal{G}}$, respectively. For more details about kernel embeddings of distributions, see Song et al. (2013) and Huang et al. (2019).

### A.5 EXTENDED HILBERT SCHMIDT INDEPENDENCE CRITERION

With the notations and results in the above section, we can calculate the dependence between $P(X)$ and $P(Y|X)$ by extended Hilbert Schmidt Independence Criterion:

$$\operatorname{HSIC}_{P(X),P(Y|X)} = \frac{1}{(N-1)^2}\operatorname{tr}\left(\mathbf{M}_X^{\mathcal{H}}H\mathbf{M}_{Y|X}^{\mathcal{G}}H\right), \tag{14}$$

where $H$ is a matrix used to center the features, whose entries $H_{ij} := \delta_{ij} - N^{-1}$. Huang et al. (2019) uses a normalized version of the estimated HSIC, which is invariant to the scale in $\mathbf{M}_X^{\mathcal{H}}$ and $\mathbf{M}_{Y|X}^{\mathcal{G}}$:

$$\begin{aligned}\operatorname{HSIC}_{P(X),P(Y|X)}^{\mathcal{N}} &= \frac{\operatorname{HSIC}_{P(X),P(Y|X)}}{\frac{1}{N-1}\operatorname{tr}\left(\mathbf{M}_X^{\mathcal{H}}H\right) \cdot \frac{1}{N-1}\operatorname{tr}\left(\mathbf{M}_{Y|X}^{\mathcal{G}}H\right)} \\ &= \frac{\operatorname{tr}\left(\mathbf{M}_X^{\mathcal{H}}H\mathbf{M}_{Y|X}^{\mathcal{G}}H\right)}{\operatorname{tr}\left(\mathbf{M}_X^{\mathcal{H}}H\right)\operatorname{tr}\left(\mathbf{M}_{Y|X}^{\mathcal{G}}H\right)}\end{aligned} \tag{15}$$

### A.6 PROOF OF THEOREM 2

**Definition 3.** *Score Equivalence. Let $D$ be a dataset consisting of $T$ records that are i.i.d samples from some distribution $P(\cdot)$. A score function $\mathbf{S}$ is score equivalent if for any two DAGs $\mathcal{G}$ and $\mathcal{G}'$ which are in the same Markov equivalence class, we have $S(\mathcal{G}', D) = S(\mathcal{G}, D)$.*

*Proof.* Let $G' = \arg\min\limits_{G} MDS_{nonlinear}(G, \mathbf{D})$, $G'' = \arg\min\limits_{G} MDS_{nonlinear}(G, \mathbf{D})$ and $G''' = \arg\min\limits_{G} MDS_{nonlinear}(G, \mathbf{D})$ be three DAGs with the same skeleton $\mathcal{S}$ as $G_1$, let $\mathbf{E}'_2$, $\mathbf{E}''_2$ and $\mathbf{E}'''_2 \in \mathbf{E}_2$ be the set of edges with cardinality $n'$, $n''$ and $n'''$, whose directions are correctly determined by $G'$, $G''$ and $G'''$ respectively.

- If $G'$ is not in the same equivalent class as $G_0$ and $n' = n_2$. According to *score consistency* of BIC, the first term in $MDS_{linear}(G', \mathbf{D})$ is larger than that in $MDS_{linear}(G^*, \mathbf{D})$ and the last term in $MDS_{linear}(G', \mathbf{D})$ and $MDS_{linear}(G^*, \mathbf{D})$ is same because $n' = n_2$, then $MDS_{linear}(G', \mathbf{D}) > MDS_{linear}(G^*, \mathbf{D})$.

- If $G'$ is in the same equivalent class as $G_0$ and $n' < n_2$. According to *score equivalence* of BIC, the first term in $MDS_{linear}(G', \mathbf{D})$ and $MDS_{linear}(G^*, \mathbf{D})$ is same and the last term in $MDS_{linear}(G', \mathbf{D})$ is larger than that in $MDS_{linear}(G^*, \mathbf{D})$ because $n' < n_2$, then $MDS_{linear}(G', \mathbf{D}) > MDS_{linear}(G^*, \mathbf{D})$.

- If $G'$ is not in the same equivalent class as $G_0$ and $n' < n_2$. then $MDS_{linear}(G', \mathbf{D}) > MDS_{linear}(G^*, \mathbf{D})$ clearly holds according to the above two cases.

All the above three cases are contradict with the condition that these three graphs are DAGs that minimize $MDS_{nonlinear}(G, \mathbf{D})$. □

### A.7 PROOF OF THEOREM 3

The conclusion that $0 \leqslant n^* \leqslant n_2$ holds clearly. The conclusion that $G^*$ is in the same equivalent class as $G_0$ can be proved similar to the proof of Theorem 2. $MDS_{nonlinear}$ is not able to guarantee $n^* = n_2$ mainly because GS is not score equivalent.

### A.8 EXPERIMENTS OF GREEDY EQUIVALENCE SEARCH

The proposed MDSS can be readily incorporated into off-the-shelf search strategies. In the main paper, we adopt policy-gradient-based search strategy (Zhu & Chen, 2019) to search the optimal causal structure. In this section, we demonstrate that greedy equivalence search (Chickering, 2002) can also be utilized as the search strategy.

Similar to the two-stage search in our main paper, we first perform greedy equivalence search on the augmented graphs (*i.e.* graphs with domain index as an additional node) to optimize the score (BIC for linear systems and GS for nonlinear systems). The output of this step is an equivalence class of the augmented graph. Then we utilize distribution shifts of the nonstationary variables to detect more edge directions. Consider the same setting as in Section 3.2 (*i.e.* , $n = 6$ and $d = 10$). In linear case, TPR, TNR, SHD for MDSS with greedy equivalence search is $0.67 \pm 0.15$, $0.75 \pm 0.16$ and $4.75 \pm 2.05$ respectively. In nonlinear case, TPR, TNR, SHD for MDSS with greedy equivalence search is $0.69 \pm 0.13$, $0.63 \pm 0.21$ and $5.80 \pm 2.67$ respectively. Compared with the results in Table 1, 1) MDSS with greedy equivalence search outperforms MC, IB and CD-NOD in both linear and nonlinear cases, 2) although MDSS with greedy equivalence search is not as good as MDSS with policy-gradient-based search in linear case, it achieves comparable results as policy-gradient-based one in nonlinear case.

### A.9 A TOY EXAMPLE

The example is consisted of 10 linear datasets with 4 variables, whose underlying causal graph is given in Figure 4(a). We use linear SEM $V_i = w_i PA_i + b_i + \epsilon_i$ to generate the data. For each variable $V_i$, $w_i$ is fixed to 1 and $\epsilon_i$ is also fixed as a standard Gaussian noise across all datasets. To introduce distribution shifts, we vary $b_i$ across datasets, here $b_3$ is chosen to be invariant (*i.e.* $V_3$ is stationary).

We first run RL on single dataset (randomly chosen from 10 datasets) and the pooling of all datasets respectively, with results shown in Figure 4(b) and 4(c). RL misidentifies the direction between $V_1$ and $V_2$ in both cases, which is reasonable because RL uses BIC plus an acyclicity constraint as score function, and BIC is *score equivalent*. Furthermore, when multiple datasets with distribution shifts are used, RL outputs erroneous edges (*i.e.* edges between $V_1$, $V_4$ and $V_2$, $V_4$) due to confounders. Next we run MDSS on the pooling of all datasets, with result shown in Figure 4(d). MDSS correctly detects variables with changing causal modules. It recovers all the directions between those 4 variables correctly as well.

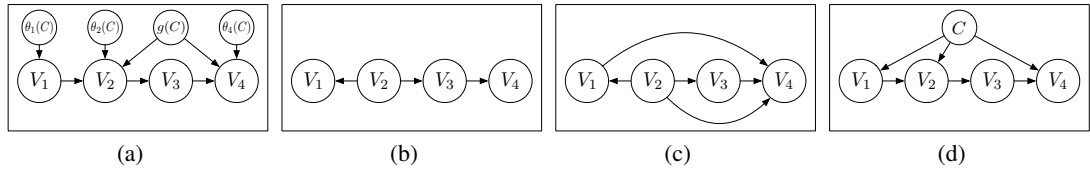

(a)           (b)           (c)           (d)

Figure 4: (a) True causal graph for the 4-variable toy example (b) Graph from RL running on single dataset. (c) Graph from RL running on multiple datasets. (d) Graph from MDSS.

## A.10    GROUND TRUE CAUSAL GRAPH FOR REAL DATA

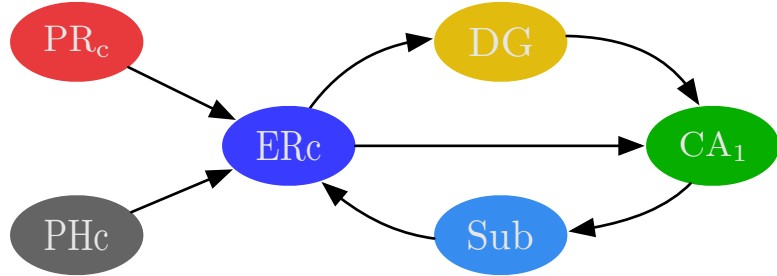

Figure 5: Ground true causal graph for real data

