# OpenReview forum: "Score-based Causal Discovery from Heterogeneous Data"
_ICLR.cc/2021/Conference — Reject_

### Official Review · AnonReviewer4 · 2020-10-27
**The paper is well-written and proposes novel causal discovery criteria. The results are rather incremental.**

**Rating:** 6
**Confidence:** 4

**Review:**

Summary of the paper: The paper introduces a novel score-based approach for causal discovery from heterogeneous data. The novel score is called Multiple-Domain Score, and can be incorporated into existing search strategies. It is shown that the MDS is guaranteed to find the correct graph skeleton asymptotically.

Strengths: The paper is rather well written. The related work is properly addressed. The proposed criteria are interesting.

Weaknesses: The results are quire incremental. The paper takes a lot from Huang et al., 2019. Assumptions 1 and 2 are somewhat obvious. The paper discusses criteria based on covariance and independence tests, however, their relation to the causal inference is not so straightforward. Algorithm 2 and Section 2.4 need more details.

Questions: The authors tell that the proposed criterion can identify more causal directions compared to the state-of-the art. I did not really find any confirmation of this statement.

Section 2.3: Why do we need $\theta$? I see that it is also the formulation adopted by Huang et al., 2019, but why the 'augmented' graph is needed? Why $\theta$ are separate nodes if they are (eq. 2 and 3) just parameters of distributions?

Section 2.3: "the dependence can be described with covariance": I agree but what about causality?

The criteria eq. 3 and 4 contain several terms. I wonder, what is the performance (and whether it is very different) of the criteria containing the last terms (the results of the covariance and HSIC) only. What is the impact of other terms?

Probably I missed it: what is h (what operation applied to the adjacency matrix)?

How the policy-gradient-based search integrates the MDS? (Section 2.4)

In experiments: is a CPDAG a causal graph? Is it always so?
The same for the Bayesian networks (the hill-climbing mentioned in section 2.4). Can we consider that a BN is a causal network? We need at least some assumptions.

I also doubt that a reinforcement learning algorithm without necessary assumptions can be considered as a causal discovery method.

---

> ### Author Response · Authors · 2020-11-18
> **Reply to R#4**
>
> We appreciate for the comments.
>
> 1. Identify more causal directions.
> 	- We use True Negative Rate (TNR, the higher the better), True Positive Rate (TPR, the higher, the better), and Structural Hamming Distance (SHD, i.e., the smallest number of edge additions, deletions, and reversals to convert the estimated graph into the true DAG, the lower the better) as criteria to evaluate the performance of the algorithms. As you can see in the experiments, our approach outperforms the others in most cases regards to these criteria, which we believe is a solid confirmation that the proposed algorithm is able to identify more causal directions.
>
> 2. Questions regards to $\theta$.
> 	- We use $\theta_i(C)$ to represent the effective parameters in the causal module of $V_i $ that are assumed to be functions of $C$， which is a random variable representing domain index. In this case, $\theta_i(C)$ also can be regarded as a random variable. That is why $\theta_i(C)$ are represented as separate nodes in augmented graphs. Augmented graphs are defined over nodes $V∪g(C)∪\theta(C)$. In practice, $g(C)∪\theta(C)$ are not directly observed, so we take $C$ as a surrogate variable and represent augmented graphs as those over $V∪C$. Algorithm 1 is based on those simplified augmented graphs. In this respect, the augmented graphs are necessary.
>
> 3. The dependence can be described with covariance.
> 	- Let's jump to part 2) $V_l \in V_C $ in Section 2.3. Take the two variables (which are named as $V_1 $ and $V_2 $) case as an example. Assume the true causal graph is $V_1→V_2 $. Causal discovery algorithms may output two possible directions: 1) $V_1→V_2 $, we name the corresponding parameters as $\theta_1$ and $\theta_2$; 2) $V_1←V_2 $, we name the corresponding parameters as $\theta_1'$ and $\theta_2'$. Obviously, case 1) represents the true causal graph so $\theta_1$ and $\theta_2$ are independent. While case 2) represents a wrong causal graph so $\theta_1'$ and $\theta_2'$ are dependent. As a result, we can use covariance as a criterion to select the true direction in linear systems (i.e., select the direction whose parameters $\theta$ have smaller covariance).
>
> 4. The terms in Equation 3 and 4.
> 	- If the criteria in Equation 3 and 4 contain the last terms (the results of the covariance and HSIC) only, there might be "degeneration issue". For more details, please refer to Degeneration issue part in Section 2.3. As for the other terms, the first one in Equation 3 and 4 can be regarded as likelihood, which is widely used in score-based causal discovery algorithms to identify edges and their directions. The second and third ones are regularization terms to avoid cycles (because we assume all the graphs are DAGs).
>
> 5. Function $h(∙)$.
> 	- Please refer to Equation 5 in Appendix A.2.
>
> 6. How the policy-gradient-based search integrates the MDS.
> 	- Original policy-gradient-based search [1] uses BIC as a score function to search the graphs. We replace BIC with MDS in MDSS (Algorithm 2). We will add more details about gradient-policy-based search in Section 2.4 in the rebuttal revision.
>
> 6. CPDAGs, BNs and causal graphs.
> 	- In causal discovery and causal inference, it is common to represent causal relations using causal graphs. A causal graph is usually directed because we need to encode causal directions. When we assume the underlying causal graph is acyclic, we denote this type of graphs as directed acyclic graphs (DAGs). A completed partially directed acyclic graph (CPDAG) is an equivalent class of DAGs. CPDAG is usually used to represent the output of some single domain causal discovery algorithms such as the PC algorithm. Bayesian Networks (BN) is a pair of DAG and a probability distribution, where the probability distribution satisfies the Markov condition for the DAG.
>
> 7. Reinforcement learning and causal discovery.
> 	- In the first paragraph of Section 1, the description of the last sentence might be kind of misleading. To be more precise, [1] does not consider a reinforcement learning algorithm as a causal discovery algorithm directly. It takes the idea from reinforcement learning: policy-gradient methods in reinforcement learning search for a best policy by receiving the stream of reward signals. (Score-based) causal discovery does a similar thing: a causal discovery algorithm searches for a causal graph that describes the observed data best by optimizing the pre-defined score function.
>
> [1] Zhu, Shengyu, Ignavier Ng, and Zhitang Chen. "Causal discovery with reinforcement learning." arXiv preprint arXiv:1906.04477 (2019).

---

### Official Review · AnonReviewer3 · 2020-10-28

**Rating:** 5
**Confidence:** 4

**Review:**

Strengths:

1. This paper studies an interesting problem of causal discovery from heterogeneous data, and does a comprehensive survey of related literature.

2. This paper proposes a novel Multiple-Domain Score Search algorithm for score-based causal discovery from different domains.

3. This paper conducts experiments on both synthetic and real-world data with comparison to state-of-the-art baselines, and the results seem promising.

Weaknesses:

1. The difficulty of causal discovery in multiple domains is not clearly shown in the experiments.

2. Some parts of the writing can be improved.

3. There might still be some important aspects of the method left to evaluate.

This paper studies a problem of causal discovery from heterogeneous data, the problem is well-motivated, and the authors give a comprehensive survey of the related work. This paper proposes a novel algorithm Multiple-Domain Score Search (MDSS) to address the problem, which searches over the space of augmented graphs to avoid the spurious edges, characterizes distribution shift with the additional domain index, and uses a novel Multiple-Domain Score (MDS) to identify causal directions. The paper evaluates the proposed method on synthetic and real-world datasets, it achieves better performance than baselines.

Here are some concerns about this paper:

1. In the experiment, it is not very clear how difficult it is for causal discovery in multiple domains (e.g. the number of spurious edges, the difference of data distributions in multiple domains, especially in the real-world dataset), which brings challenges to convincingly evaluate the superiority of the proposed method.

2. As for the writing, some parts of the paper are not very easy to follow for non-experts. It would be better if the terms (e.g., causal sufficiency, causal modules) can be briefly introduced when first mentioned.

3. Also, I think there may be many interesting aspects to evaluate in the experiments, e.g., how robust the algorithm is under different levels of domain shift; how each component in the method contributes to the performance (some ablation studies may be considered).

---

> ### Author Response · Authors · 2020-11-18
> **Reply to R#3**
>
> Thanks for the great suggestions/comments.
>
> 1. The difficulty of causal discovery in multiple domains.
> 	- We have compared the performance of the proposed algorithm with those designed for single domain in our experiments. In Table 2, we evaluate the algorithms designed for single domain on both single-domain data and multiple-domain data (i.e. the values in column "single" and "pool"). The results show that algorithms designed for single domain have difficulty on multiple-domain data. Specifically, despite the minor increase in TPR, their TNR decrease dramatically when data from all domains are used. This phenomenon proves that applying algorithms designed for single-domain on multiple-domain data will introduce spurious edges. As a comparison, we evaluate the proposed MDSS on multiple-domain data. As one can see, MDSS fits the multiple-domain data much better.
>
> 2. Writing improvement.
> 	- Thanks for the advice. Intuitively speaking, causal sufficiency means that all common causes (i.e., parents) of variables are measured. The explanation of causal modules is available in the second paragraph of Section 2.1. We will inspect our paper and add brief introductions to these terms in the rebuttal revision.
>
> 3. How robust the algorithm is?
> 	- We allow that the mechanisms and parameters associated with the causal model change across data sets or over time, or even vanish or appear in some domains, while we assume that causal directions do not flip and variable set are fixed across domains. Considering the changing causal directions and number of variables is definitely interesting, and our method has paved the way for addressing these more challenging problems.
>
> 4. How each component in the method contributes to the performance?
> 	- The proposed method contains two components: skeleton estimation (Algorithm 1) and MDS search (Algorithm 2). Since MDS search relays on the output of skeleton estimation, we only conduct ablation study to see how MDS search improves the performance. In other words, we keep the directions in step 3 of Algorithm 1 and output $G_1 $ (Algorithm 2 is not executed). We use the same experimental setting as Table 1. The results (TPR, TNR and SHD) are 0.58±0.21, 0.92±0.07 and 7.95±4.51 for linear data, 0.32±0.22, 0.58±0.22 and  6.20±0.26 for nonlinear data. Readers can compare the results with those of MDSS in Table 1. It is obvious that MDS search identifies more directions and boost the performance.

---

### Official Review · AnonReviewer2 · 2020-10-28
**The paper contains mistakes**

**Rating:** 3
**Confidence:** 4

**Review:**

The paper presents a score-based approach for causal graph discovery from heterogeneous data. The approach is claimed to be guaranteed to find the correct graph skeleton asymptotically, and can detect more causal directions than previous algorithms designed for single domain data.

Pros:
-The paper addresses an important problem of causal discovery from heterogeneous data.

Cons:
I have doubts about the correctness of many conclusions in the paper. For example, it looks to me that for Theorem 1 to be true, the pooled data $D_C$ must be i.i.d. samples from distribution P(V,C). However, $C$ is defined as the domain index. The problem setup simply assumes we are given $n$ data sets from $n$ domains. Then it’s natural to assume the domain index “C” will stand for 1, 2,  …, n. Under this setting, the pooled data are not i.i.d. samples, Theorem 1 will not hold, and I believe some other conclusions in the paper will not hold either. For Theorem 1 (and other conclusions) to hold, I believe one has to assume the domain index C is a random variable, such that the data sets from different domains are drawn randomly based on some P(C) and P(V|C). However, this may not be a realistic setting, and is not how the problem is set up in the paper. I don’t think the real data sets in the experiments satisfy this setting (not clear how the synthetic data sets are generated).

Overall, I think the paper has mistakes and vote for reject.

---

> ### Author Response · Authors · 2020-11-18
> **Reply to R#2**
>
> We thank the reviewer for the comments.
>
> We believe that the related conclusions in our paper are correct, because the joint distribution $P(V,C)$ is fixed, and thus the pooled data (including both values of $V$ and $C$) are i.i.d. samples from distribution $P(V,C)$. **Note that it is the $P(V|C)$ that shifts across domains**. That is the reason why we explicitly include $C$ into the causal system. Please see the detailed explanations below.
>
> When $C$ is the domain index, $P(C)$ follows a discrete uniform distribution whose possible values are $1,2,…,n$. Correspondingly, the generating process of heterogeneous data can be considered as follows: First we generate random values from $P(C)$, and then we generate data points over $V$ according to the SEM in Equation 1. Finally, generated data points are sorted in ascending order according to the values of $C$ (i.e., data points having the same value of $C$ are regarded as belonging to the same domain). In other words, we observe the distribution $P(V│C)$, where $P(V│C)$ may change across different values of $C$, resulting in distribution-shifted data. Note that if we do not include $C$ into the system explicitly, samples of $V$ are not i.i.d. However, after explicitly including the domain index $C$ into the system, $P(V,C)$ is fixed, and thus the pooled data are i.i.d. samples from distribution $P(V,C)$.
>
> Moreover, you may consider it from another perspective: the task of classification, where we have data $X$ and class labels $C∈{1,2,3,…}$. In classification, the labels are often regarded as samples from a discrete random variable. And the joint data $(X_1,C_1 ),(X_2,C_2 ),…$ are regarded as i.i.d. samples from joint distribution $P(X,C)$. Therefore, the domain index can be considered as a label in our paper.

---

> > ### Comment · AnonReviewer2 · 2020-11-22
> > **more questions**
> >
> > Thanks for the clarification of the problem setup. However, it looks to me Theorem 1 still does not hold, specifically the "if" part. Assume the edge between  $V_i$ and $V_j$ does not exist in the true graph $G_0$, say the true graph is $V_i \rightarrow X \rightarrow V_j$. Let $G_C$ be $V_i \ X \ V_j$ (no edges), and $G'_C$ be $V_i \rightarrow V_j, X$. We have $Pa_j$ in $G_C$ is empty set. $V_j$ is NOT independent of $V_i$ given empty set according to the true graph, therefore S(G', D) > S(G,D).

---

> > > ### Author Response · Authors · 2020-11-23
> > > **Reply to the question**
> > >
> > > According to your example, the true graph $G_0$ is $V_i⟶X⟶V_j$, $G_C$ is $V_i\quad X\quad V_j$, and $G_C^{\prime}$ is $V_i⟶V_j\quad X$. It is true that $S(G_C^{\prime},D)>S(G_C,D)$ according to local consistency. However, this is only a local step. Let us show what will happen if you proceed with this example using some score-based algorithms (e.g. GES[1]). Step 1, according to your example, the arrow $V_i⟶V_j$ is added to the empty graph, resulting $V_i⟶V_j\quad X$. Step 2, either $V_i⟶X$ or $X⟶V_j$ can be added according to Theorem 1. Say we add $V_i⟶X$, resulting $V_j⟵V_i⟶X$. Step 3, $X⟶V_j$ can be added according to Theorem 1, resulting $V_i⟶X⟶V_j$ and also $V_i⟶V_j$ (we split the graph because we are not able to plot such a graph here). Step 4, $V_i⟶V_j$ can be deleted according to Theorem 1, resulting the true graph $V_i⟶X⟶V_j$. As a result, Theorem 1 holds in this example. As stated in our paper, The local consistency criteria has been proven to hold on BIC and GS score [2], which were used to search for causal graph on single domain data.
> > >
> > > [1] Chickering, David Maxwell. "Optimal structure identification with greedy search." Journal of machine learning research 3, no. Nov (2002): 507-554.
> > >
> > > [2] Huang, Biwei, Kun Zhang, Yizhu Lin, Bernhard Schölkopf, and Clark Glymour. "Generalized score functions for causal discovery." In Proceedings of the 24th ACM SIGKDD International Conference on Knowledge Discovery & Data Mining, pp. 1551-1560. 2018.

---

> > > > ### Comment · AnonReviewer2 · 2020-11-23
> > > > **reply**
> > > >
> > > > I'm not saying GES does not work. This example just shows that the "if" part of Theorem 1 does not hold: we have S(G',D) > S(G, D), but no edge between $V_i$ and $V_j$ in the true graph.

---

> > > > > ### Author Response · Authors · 2020-11-25
> > > > > **Reply about Theorem 1**
> > > > >
> > > > > ˙Thanks for pointing out this problem. We have corrected the theorem and its proof in the updated version. Theorem 1 follows from the optimality of GES on a single dataset, except that we add the domain index variable $C$ in the algorithm.

---

> > ### Comment · AnonReviewer2 · 2020-11-23
> > **more comments**
> >
> > The first sentence in Section 2.3. “For each variable $V_k \in  V_C$, we prove that it is possible to determine the directions of edges that connect to $V_k$.”
> >
> > Do you have a formal proof of this? It looks to me, for this to be correct, some assumptions need to be made on the types of data available. The arguments for case 1) are very informal. In fact, those claims ignore other variables and are only correct assuming the only variables are $C, V_k, V_l$. The arguments for case 2) look more heuristics than a formal result.
> >
> > I find the descriptions for case 2) vague and not clear. Can you explain the following?
> >
> > -What does “$\theta_1$ and $\theta_2$ are independent” mean?
> >
> > -“For linear systems, the dependence can be described with covariance. θ1 and θ2  can be easily obtained by regressing V1 and V2 on their parents respectively.”
> >
> > What are θ1 and θ2 when V1 and V2 have multiple parents? That the relations among $V_i$ are linear does not mean the relation between θ1 and θ2 are linear.
> >
> > -This notation $V_C^k$’s  “parents $PA^k_C \in V_C$" confuses me. First, is $PA^k_C$ a set or a single variable? If it’s a single variable, should you consider all of them? Second, what about the parents of $V_C^k$ that are not in $V_C$? What do you mean by “the parameters of $V_C^k$  or $PA^k_C$”?
> >
> > -When $V_C^k$ has multiple parents X, Y, etc., it seems instead of considering $P(V_C^k| X, Y, ...)$, only $P(V_C^k| X)$ is computed. Why should $P(V_C^k| X)$ be independent of $P(X)$ when $X$ may influence $Y$?
> >
> > I’m not sure the claim regarding the confounding case in Figure 1(d) makes sense. The dependence from the confounder and that caused by the wrong direction could also cancel out each other.

---

> > > ### Author Response · Authors · 2020-11-23
> > > **Reply to the comments**
> > >
> > > Thanks for your additional comments.
> > >
> > > In the first case, we make use of the invariance property of a causal system to find the causal directions. This property has been used in [1] to determine edge directions by extending the PC algorithm. In our case, after incorporating domain index $C$ in score-based search method such as GES, our method is guaranteed to find the causal direction due to the global and local consistency of the scores like BIC and GS score. The proof of this part is trivial because the score-based method is guaranteed to find the Markov equivalent class for augmented graphs including the domain index $C$. That is to say, it can distinguish between the V structure $C⟶V_k⟵V_l$, where $C$ and $V_l$ are independent conditioned on an empty set, and the structures $C⟶V_k⟶V_l$, $C⟵V_k⟶V_l$, $C⟵V_k⟵V_l$, where $C$ and $V_l$ are independent given $V_k$. Since $C$ is domain surrogate variable which, we have $C⟶V_k$, thus the algorithm will output either $C⟶V_k⟵V_l$ or $C⟶V_k⟶V_l$. Thus, the causal direction between $V_k$  and $V_l$ can be determined.
> > >
> > > In the second case, we make use of the independence change property of a causal model [2,3,1]. That is, the parameters in the true causal model change independently while the parameters in a wrong causal model will change dependently. In this case, we use the dependence of parameters of each non-stationary causal module as a score, which is the last term in Eq (3) and (4).
> > >
> > > 1. What does “$θ_1$ and $θ_2$ are independent”mean?
> > > 	- We have dropped the notation of domain index $C$ here for simplicity. To be more precise, $θ_1$ and $θ_2$ should be written as $θ_1(C)$ and $θ_2(C)$ as in Figure 1(b). Here the domain index $C$ is a random variable. $θ_1(C)$ and $θ_2(C)$ are mappings of $C$, which can also be regarded as random variables. By saying “$θ_1$ and $θ_2$ are independent”, we mean the causal module of $V_1$ and the causal module of $V_2$ change independently across domains. Suppose there are n domains, we will get a sample of size n for $θ_1$ and $θ_2$ and thus we can measure the dependence of $θ_1$ and $θ_2$.
> > >
> > > 2. What are $θ_1$ and $θ_2$ when $V_1$ and $V_2$ have multiple parents? That the relations among $V_i$ are linear does not mean the relation between $θ_1$ and $θ_2$ are linear.
> > > 	 - When $V_1$ and $V_2$ have multiple parents, $θ_1$ and $θ_2$ are random vectors. For linear systems, the random vector is composed of all parameters in the regression equation.
> > > 	- Here by saying “linear systems”, we simply assume all the relations including those between $θ_1$ and $θ_2$ are linear. This is a quite ideal assumption but making the idea behind case 2 easy to follow. We also consider the non-linear system and learning objective is in Eq (4).
> > >
> > > 3. Questions about $PA_C^k$
> > > 	- We apology that “the parameters of $V_C^k$ or $PA_C^k$” is a confusing expression. It should be “the parameters of $V_C^k$’s or $PA_C^k$’s causal modules”. We will modify it in the rebuttal version.
> > > 	- $PA_C^k$ is a set of variables. Precisely speaking, “the dependence between parameters of $V_C^k$’s causal module and the parameters of $PA_C^k$’s causal modules” should be "the sum of the dependence between parameters of $V_C^k$’s causal module and the parameters of the causal module of each variable in $PA_C^k$". The notation $PA_C^k∈V_C$ (which should be $PA_C^k⊆V_C$) is confusing, we will modify it in the rebuttal version.
> > > 	- There is no need to consider the parents of $V_C^k$ that are not in $V_C$. Because this belongs to case 1 (i.e. $V_C^k$ is connected to $C$ but the parent is not).
> > >
> > > 4. When $V_C^k$ has multiple parents $X$, $Y$, etc., it seems instead of considering $P(V_C^k | X,Y,...)$, only $P(V_C^k | X)$ is computed. Why should $P(V_C^k | X)$  be independent of $P(X)$  when $X$ may influence $V_C^k$?
> > > 	- No, we compute the causal module of $V_C^k$, which is conditional on all the parents of $V_C^k$.
> > >
> > > 5. Confounders cancel the dependence
> > > 	- Here we assume the confounder is not strong, such that its effects do not change the final results. When the confounder is strong, it is possible that the dependence between parameters will be smaller in the reverse causal direction.
> > >
> > > [1] Huang, Biwei, Kun Zhang, Jiji Zhang, Joseph Ramsey, Ruben Sanchez-Romero, Clark Glymour, and Bernhard Schölkopf. "Causal discovery from heterogeneous/nonstationary data." Journal of Machine Learning Research 21, no. 89 (2020): 1-53.
> > >
> > > [2] Pearl, Judea. "Models, reasoning and inference." Cambridge, UK: CambridgeUniversityPress (2000).
> > >
> > > [3] Tian, Jin, and Judea Pearl. "Causal discovery from changes." arXiv preprint arXiv:1301.2312 (2013).

---

### Official Review · AnonReviewer5 · 2020-11-07
**Review for Score-based Causal Discovery from Heterogeneous Data**

**Rating:** 7
**Confidence:** 4

**Review:**

This paper proposes strategies for learning the structure of multiple sets of data observed over a common set of variables which may exhibit distribution shift. The authors address this problem by augmenting the dataset with an indicator variable which indicates membership to  dataset. After augmenting the dataset standard algorithms for structure learning are applied, with the additional restriction that the indicator variable may only be an ancestor. The authors provide theory that shows the procedure consistently estimates the local structures. The authors then show how the additional information obtained from the structure learned with the context variable can be used to disambiguate directions. Experimental results show the efficacy of the proposed approach.

Overall, I think this is a sensible idea and contains some nice results. I do have a couple of questions:

(1) In the paper the authors explicitly limit the algorithm to the case where all domains observe the same variables, however it seems like this need not be the case?
(2) The authors should cite the literature on learning with mixtures of Bayesian networks. While the aims are different, the model representation is very similar.
(3) Can the authors provide intuition on whether this degeneracy will also become an issue if there are errors in the skeleton detection algorithm?

---

> ### Author Response · Authors · 2020-11-18
> **Reply to R#1**
>
> We thank the reviewer for the great suggestions.
>
> 1. All domains observe the same variables.
> 	- In this paper, we assume that all the domains observe the same variables and focus on addressing the distribution shift problem. There are some existing works on the non-identical variable sets problem, but they cannot handle the case when the causal edges change across domains [1][2]. In the presence of both distribution shift and non-identical variables, the problem will become even harder, because it is hard to estimate the distribution of those non-overlapping variables. We believe that this problem is worth investigating and will consider it as our future work.
>
> 2. Literature on learning with mixtures of Bayesian networks.
> 	- Related literature will be added into the rebuttal revision, and we will emphasize the differences.
>
> 3. Degeneracy.
> 	- Errors in the skeleton detection algorithm will not lead to degeneration issue. As we discussed in Part "Degeneration issue" in Section 2.3, degeneration issue means that the MDS penalty tends to eliminate any edges between each $V_C^k$ and its parents if we apply search strategies over the entire space of graphs over $V$ to optimize MDS. Errors in the skeleton may result in erroneous edges (i.e., missing edges, redundant edges, or edges with wrong directions) in the estimated causal graph, but will not lead to degeneration issue as long as we limit the search space of graphs over $V$ within that defined by the skeleton from Algorithm 1 to optimize MDS (i.e., Algorithm 2 is based on the fixed skeleton from Algorithm 1).
>
> [1] Triantafillou, Sofia, Ioannis Tsamardinos, and Ioannis Tollis. "Learning causal structure from overlapping variable sets." In Proceedings of the Thirteenth International Conference on Artificial Intelligence and Statistics, pp. 860-867. 2010.
> [2] Huang, Biwei, Kun Zhang, Mingming Gong, and Clark Glymour. "Causal Discovery from Multiple Data Sets with Non-Identical Variable Sets." In AAAI, pp. 10153-10161. 2020.

---

### Author Response · Authors · 2020-11-21
**Revised version uploaded**

We have uploaded a revised version of the paper, following the suggestions and comments from the reviewers. The changes including:

1. In Section 1, we add some literature about mixture of Bayesion networks, following R#1's suggestions.
2. In Section 1, we add brief introduction to "causal sufficiency", following R#3's suggestions.
3. In Section 2.2, we add some descriptions about how domain shifted data are sampled, and why the pooled data with domain index are i.i.d. samples from $P(V, C)$, following R#2's comments.
4. In Section 2.2, we correct Theorem 1 and the corresponding proof in Appendix A.1, following R#2's comments.
5. In Section 2.4 , we add some details about causal discovery with reinforcement learning and how policy-gradient-based search integrates the MDS, following R#4's suggestions.
6. In Section 3.2, we add the results of some abalation studies to validate that MDS search contributes to the performance, following R#3's suggestions.

Once again we thank all the reviewers for their effort and helpful comments/suggestions.

---

### Decision · Program_Chairs · 2021-01-07
**Final Decision**

**Decision:**

Reject

**Comment:**

This submission tackles an important problem and presents interesting ideas. I am confident that the research will lead to good publications. However, in the particular situation here, AnonReviewer2 had serious concerns that are shared by me. The authors made a great effort to clarify the situation, but the current situation still leaves me uncertain about the presentation and correctness of everything. Because some issues were major, it is not easy to re-evaluate and take new conclusions in the short time of this process. I hope the authors do not take this too negatively, but given all the comments and discussions, it is best that another round of improvements and reviews be conducted.